# "You need to dispose of them somewhere safe": Covid-19, masks, and the pit latrine in Malawi and South Africa

Marc Kalina[1,2]⊛*, Jonathan Kwangulero[3‡], Fathima Ali[2‡], Elizabeth Tilley[1]⊛

**1** Department of Mechanical and Process Engineering (MAVT), ETH Zürich, Zürich, Switzerland, **2** School of Engineering, University of KwaZulu-Natal, Durban, South Africa, **3** Department of Environmental Health, University of Malawi, The Polytechnic, Blantyre, Malawi

⊛ These authors contributed equally to this work.
‡ These authors also contributed equally to this work
* mkalina@ethz.ch

**Data Availability Statement:** All underlying data from this study is publicly available at https://figshare.com/s/b9400844fed2c938240e.

## Abstract

The ongoing Covid-19 pandemic has generated an immense amount of potentially infectious waste, primarily face masks, which require rapid and sanitary disposal in order to mitigate the spread of the disease. Yet, within Africa, large segments of the population lack access to reliable municipal solid waste management (SWM) services, both complicating the disposal of hazardous waste, and public health efforts. Drawing on extensive qualitative fieldwork, including 96 semi-structured interviews, across four different low-income communities in Blantyre, Malawi and Durban, South Africa, the purpose of this article is to respond to a qualitative gap on mask disposal behaviours, particularly from within low-income and African contexts. Specifically, our purpose was to understand what behaviours have arisen over the past year, across the two disparate national contexts, and how they have been influenced by individual risk perceptions, established traditional practice, state communication, and other media sources. Findings suggest that the wearing of cloth masks simplifies disposal, as cloth masks can (with washing) be reused continuously. However, in communities where disposable masks are more prevalent, primarily within Blantyre, the pit latrine had been adopted as the most common space for 'safe' disposal for a used mask. We argue that this is not a new behaviour, however, and that the pit latrine was *already* an essential part of many low-income households SWM systems, and that within the Global South, the pit latrine fulfils a valuable and uncounted solid waste management function, in addition to its sanitation role.

## Introduction

Throughout the Covid-19 pandemic, the widespread use of face coverings, including reusable and disposable masks, has been broadly advocated as one of the simplest, most cost effective public health measures to prevent person-to-person disease transmission, especially in contexts when effective social distancing may be difficult [1]. Given the magnitude of the crisis

**Funding:** The author(s) received no specific funding for this work.

**Competing interests:** The authors have declared that no competing interests exist.

posed by the pandemic, and the scale of disruption to global systems, public health agencies, institutions and research institutes such as the World Health Organisation (WHO) [2], were required to rapidly produce and disseminate documents, videos, and graphics on correct wear, as well as proper handling precautions [3,4], for used, and potentially contaminated masks. This urgency is rooted in the ability for contaminated masks to carry potential pathogens, including Covid-19, long after being discarded (up to 48 hours), becoming a possible point of transmission or cross-contamination [5–7].

Similarly, the academic community was quick to reflect on implications for disparate waste management systems, which must now cope with massive influxes of contaminated and hazardous medical waste, including potentially, billions of discarded face masks [8]. As the scope of the pandemic has grown, and likewise, the scale of the waste management challenge, a growing body of academic literature has emerged, focused on different aspects of disposal and end-of-life for the ubiquitous mask. Many scholars, optimistically, have centred investigations on best practice techniques for mask disposal [9–13], including a growing body of literature on possible recycling and valorisation pathways for different disposable mask typologies [14–16]. However, more commonly, academic discourse on masks disposal has been cautionary, or even alarming, warning of the risks of improper disposal broadly [4,8,17–20], including possible environmental health risks [20], and potential impacts on the environment [18,20,21], and the marine environment in particular [8,22–25]. Other contributions have assessed or forecasted the burden of growing numbers of disposed masks (and other PPE) on municipal waste management systems [21,26], including the risk to collectors of handing potentially contaminated municipal solid waste (MSW) [27]. Furthermore, a significant number of investigations have been produced from within the Global South, where municipal solid waste management (SWM) systems may be less well positioned to cope with a sudden influx of potentially contaminated waste [4,13,17,18,20,27–31].

Nevertheless, despite this recent output of research on mask disposal, there remains a paucity of investigations documenting individual disposal practices, including the perceptions and understandings that drive personal behaviours and decision-making. A few perspectives have emerged. For instance, Huynh [17], Islam et al. [28], Li et al. [30], and Scalvenzi et al. [32] have all produced quantitative assessments of public knowledge and attitudes towards mask disposal, with all but the last cited study being contextualised within the broader Global South. However, as of yet, no literature has emerged from African contexts. Moreover, there remains a larger gap for qualitative assessments of mask disposal behaviours, which can centre the socio-cultural understandings which inform them.

Likewise, there is a small, but growing awareness within water, sanitation, and hygiene (WASH) literature about the role that toilets, and pit toilets in particular, play within household solid waste management systems. Pit toilets or latrines, on which an estimated 1.77 billion people rely globally, collect and store faecal sludge onsite, and are meant to be covered over when full to allow the waste to decompose [33]. However, as Sisco et al. [33] note, there is often no space to dig a new latrine, especially in urban areas, and, as a result, emptying the pit is necessary. Early research characterised pit toilets as receptacles of convenience, decrying the burden that pit trash adds to the pit emptying process, but without seriously interrogating the motivations that drive individuals to toss their rubbish down the toilet [33,34]. However, newer scholarship has begun to unpack, what appears for many across the globe, to be a niche role for the pit toilet in the disposing of feared, dangerous, or potentially embarrassing waste items. For instance, a number of investigations have referred to the high prevalence of potentially dangerous objects found in in pits, such as hypodermic needles and broken glass [33–35]. Moreover, Roxburgh et al. [36], writing about women's menstrual health, have described how for many Malawian women, pit latrines function as vital 'emergency' disposal options for

disposing of feminine hygiene products and menstrual blood, especially in cultures where such items carry a stigma or taboo. They connect this behaviour with Foucault's [37] notion of 'heterotopias of deviation', in which menstrual blood, as an 'undesirable body' is hidden from public view to maintain a sanitised utopia. And while traditionally, pits were filled and covered at the end of their useful life, replaced with a newly dug hole every 10–20 years, dense urban life requires that the contents (a heterogeneous mix of excreta, trash, and soil) are emptied, and transported away for (hopefully) treatment. Unfortunately, even the strongest pumps on the best vacuum trucks have insufficient suction or agility to easily empty a pit full of trash. New innovations have attempted to address the challenge posed by trash in pits [33], but no commercially viable tools have come to market, and there removal of trash from pits may even cause increased exposure to the worker through pathogenic aerosols [38]. This scholarship, or rather the lack of it, raises further questions about what other waste items may qualify as 'undesirable', and are also being hidden down the world's nearly 2 billion pit latrines.

Drawing on 96 semi-structured interviews within four different low-income communities in Blantyre, Malawi and Durban, South Africa, the purpose of this article is to qualitatively investigate face mask handling and disposal behaviour within low income African communities during the Covid-19 pandemic. Specifically, we wanted to understand what practices have arisen over the past year, across disparate national contexts, and how they have been influenced by individual risk perceptions, established customs, state communication, and media sources. Results show that in Durban, where cloth masks overwhelmingly predominate, handling is straightforward, with most respondents reporting washing and reusing their masks regularly. Few had ever disposed of their masks, and those that did utilise municipal waste management services, which did not see significant disruptions during 'lockdowns'. In Blantyre, however, results were more complex, where respondents were much more likely to use disposable masks, respondents had adopted specific disposal methods, centred on the pit latrine as, the quote in the title of the article suggests, and understood 'safe' place to dispose of a feared and potentially hazardous waste item. These behaviours were informed by a mix of traditional custom, state news information, and social media, and have implications for both WASH and SWM interventions, as it sheds light on the ways in which the pit latrine may be considered by many across the Global South, *part and parcel* of the household SWM system, particularly in a time of crisis.

## Methodology

To answer the questions posed in this study, 96 semi-structured interviews were conducted across four different low-income communities in both Durban, South Africa and Blantyre, Malawi over two periods between August and October, 2020 and in March 2021. Specifically, 32 interviews were conducted with residents of Johanna Road, an informal settlement (named after the road it straddles) located on the fringes of Durban's central suburbs (Fig 1), 28 interviews were conducted with residents of Ndirande, a dense low-income neighbourhood on the outskirts of Blantyre, 18 were conducted with residents of Likhubula, a small but sprawling low-income community near Blantyre's airport (similar to Ndirande, but chosen specifically for density and mix of formal and informal housing), and a final 18 interviews were conducted with pedestrians and street vendors in Blantyre's central business district (CBD). All communities and respondents were purposively selected, based on where the researchers had previously established research connections, and could therefore easily negotiate access, and could safely conduct in-person, qualitative research during a pandemic. Both areas are extremely dense, informal, and in the centre parts, isolated.

Despite the different national contexts, all of the communities (and respondents) selected in both South Africa and Malawi are low-income, with predominantly informal housing, and

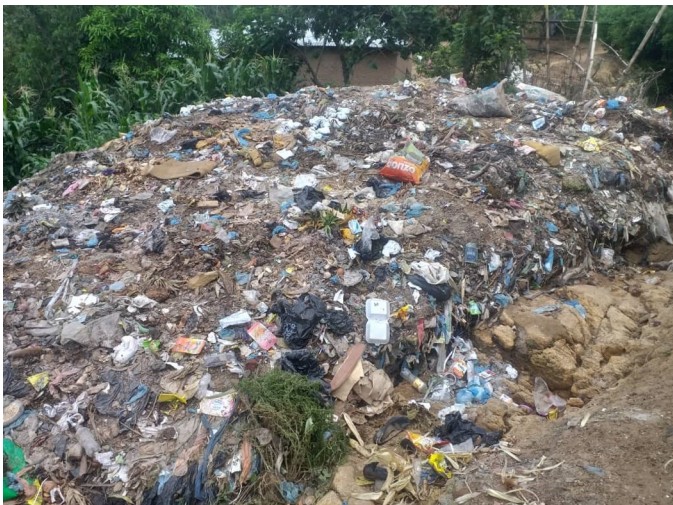

**Fig 1. One of the numerous dumping grounds in Ndirande.**

poor, or no, public services. Though an informal community, Johanna Road has better public services than the Malawian case studies, with regular municipal garbage collection, and communal water taps and ablution blocks. Neither Ndirande nor Likhubula have household municipal waste collection but rely on infrequently serviced public skips (dumpsters). Residents of Ndirande who don't use the skip, burn a portion of their household waste, and dump what cannot be burned into numerous informal dumping grounds, or into the nearby watercourse. Likewise, residents of Likhubula, also burn their household waste and dump in the nearby river, but also have an irregularly collected municipal dump point near the main road which some residents use.

Interviews were semi-structured, with questions centred on Covid-19, masking behaviours and household waste throughout the pandemic. As noted, 96 interviews were conducted, across the four communities, which was the maximum feasible number that could be conducted with the time and resources available; however, the consistency of respondents' responses suggests that saturation was achieved. Interviews were conducted in the local language (Chichewa for Blantyre, and isiZulu in Durban), audio recorded, and transcribed into English. Participation was voluntary, and responses were recorded anonymously. Oral consent was gained from each respondent before every interview. This research was approved by the National Committee On Research In The Social Sciences And Humanities in Malawi, Protocol NO. P.03/19/356. Data were analysed thematically, and stored, transcribed, and then coded within the software programme Nvivo, which organises materials and assists with the coding process.

Both Malawi and South Africa have suffered from the Covid-19 pandemic, but less severely than the stark numbers that were initially forecasted at the onset of the pandemic [39], and less than many richer, Western nations. The factors which have contributed to these successes will undoubtedly be unpacked over the coming years, yet, nonetheless, this study occurred in the midst of a pandemic, and in the midst of two different state and societal responses to Covid-19. In South Africa, in March 2020, the national government imposed the first of a series of lockdowns restricting movement, closing business, and making masking in public spaces mandatory. Although, at the time of the fieldwork, some restrictions had been lifted, public masking remained mandatory. Yet, as interviews revealed, service delivery, including waste collection, even in Durban's poorest communities, such as Johanna Road, was not severely

disrupted, even in the most restrictive lockdowns. In Malawi, a nationally imposed lockdown was blocked by the high courts in April 2020. However, a mask mandate was introduced, but it was never enforced. As one respondent from Ndirande (08/10/2020) described, "There was no change. Yes, restrictions were there, but people were not following them and the government wasn't enforcing them. Yes, civil servants and a few selected people were in lockdown, but for many people it was business as usual." Thus, aside, from sporadic mask wearing in public, Covid-19 has had a less visible or tangible impact on Malawian daily life.

## Results

### Masks and mask problematisations

Among the respondents from the four target communities, the responses were generally clustered along national lines: In Durban, regardless of the community, respondents overwhelmingly reported owning and using reusable cloth masks. Most individuals owned one or two cloth masks, and washed them frequently. As such, disposal was not a common issue, and very few individuals described their masks ever degrading to the point where it could no longer be reused, and hence needed to be disposed of. The few in Durban who did describe needing to dispose of their mask (either a no longer reusable cloth masks or the rare disposable mask), did not describe taking special precautions for disposal, and mixed the discarded mask in with their bag of MSW.

Amongst the various respondents in Blantyre, more than two-thirds of respondents reported using disposable masks, either solely, or in conjunction with a reusable, cloth mask. Although we did not specifically gather data on the typologies of disposable masks being used, anecdotal evidence suggests that basic surgical and dust masks, which would have been available prior to the pandemic, were most common.

Used masks were clearly problematised by respondents as a health risk, and as a possible vector for Covid-19. Furthermore, most individuals seemed to have a basic understanding of how a mask could become contaminated. One respondent in Likhubula (09/10/2020) explained, "Covid is very infectious and we need to take care. When you wear a mask you leave viruses there if you are positive, and if someone wears the mask definitely he or she can get sick." A few had more nuanced understandings, describing droplets and pathogens that could attach to, and linger on, used masks, making them a possible source of Covid-19 transmission.

Seen as a source of infection, improperly disposed of masks were also widely problematised, across each of the Blantyre case studies, as a public health risk, but more specifically, as a risk to area children. To these respondents, the risk of discarded masks to adults was low, because adults know that masks are potentially hazardous, and to avoid them. However, children are unaware of the risk, and may inadvertently handle a used, discarded mask and become infected. Thus, proper disposal was, for many, a duty- to safeguard their community, and protect the vulnerable, as the following quotes from respondents illustrate:

> It`s hard to get sick from a mask because it takes someone to use a used mask of a corona positive person, and a normal human wouldn't pick up a mask off the ground–but a child might, that is the risk (08/10/2020).

> If it's not managed properly [the masks]. . . children may wear them. As you know children are easily attracted to things like this. As a result, they may catch the virus because of using a used mask (08/10/2020).

> There is a risk to children, because when they see a mask they pick it up to wear, which may get them infected. . . .. This is why they [masks] must be disposed of properly, and not tossed on the ground (07/10/2020).

This fear was first targeted towards careless individuals who might discard a mask on the ground, which may be unwittingly picked up by a child. However, it was more seriously rooted in the case study communities' lack of waste management services. As previously described, both Ndirande and Likhubula lack municipal waste management services (Likhubula has serviced one dump point by the road with extremely sporadic collection), and as such, dumping is common (Fig 1). Because children play freely throughout the community, they often play within dumping grounds and play with waste objects. So for many respondents, the concern of children encountering masks was more nuanced, as were they typical disposed their household waste could not be made out of bounds to area children.

## Disposal

Respondents in Blantyre adopted two main disposal methods for used masks, either burning them or disposing them down their pit latrines. A small minority (4) reported disposing masks along with their household waste. However, more than 90 percent of Blantyre respondents (61) reported using a pit latrine to dispose of their mask, at least some of the time. While, about a third (18) reported either burning their masks exclusively, or a mix of both burning and disposing down pit latrines. Burning was believed to be the most effective disposal method as it would eliminate any risk of infection. However, starting a fire is both time and labour intensive, and many individuals were wary of holding onto their masks, once used, so although a few describe starting fires specifically to burn a mask, or burning masks along with their general household mask, most respondents admitted to often disposing of their masks down a pit out of convenience, as a resident of Ndirande (07/10/2020) articulated, "there are two ways to [get rid of a mask]. We either burn them or dispose of them in the pit latrine. But, I prefer dropping them into the pit latrine, because it's much easier."

Pit latrines were considered an ideal place to dispose of a mask, because once dropped down the pit, the mask would remain there, and hence nobody would come into contact with the discarded (and feared) item. As a respondent in Blantyre CBD (09/10/2020) succinctly described, "I dump in pit latrines so that no one will be able to interact with [the waste]." As previously noted, many participants articulated concern about children coming into contact with discarded (and possibly infectious) masks on the streets. These fears influenced individuals' disposal choices, as a pit latrine was thought to be a safe space for disposal. A different respondent in the Blantyre CBD described this decision-making:

*We dispose of masks in pit latrines because in our community there is a possibility that children may pick and play with masks if they are disposed of in other ways. So, to protect children from Corona we dump in the pit latrine or burn (10/03/2021).*

Others expressed concern about the safety of even burying contaminated masks, for fear that they might be exhumed inadvertently. As such, masks were clearly feared by respondents, and the pit latrine was considered the one reliable place you could toss one, and it would not come back to threaten your health or that of your community.

Finally, respondents who disposed of masks down pit latrines were all generally aware that tossing solid waste down the pit would fill it up and shorten its life, but considered it worth it. A minority of the respondents have their pits, when full, professionally emptied. A few of these individuals did express concern that the disposed masks may slow down and consequently, increase the price of the pit emptying process. However, as one such respondent in Likhubula (10/03/2021) remarked, "still, we don't feel it's serious enough to change how we use or manage our pit latrine." Moreover, none of these individuals expressed any concern about the possible risk of

exposure to the pit emptiers from the disposed masks. Several others described using a chemical additive, available in local markets, which they pour down the pit and which, over the period of a month (during which time they construct a temporary latrine), breaks down faecal matter and clears space within the pit, extending its life. Most were unsure how the additive would respond to the masks. However, the majority of respondents do not empty their pit latrines, but have enough space to seal their existing pit when it fills and dig a new one. For these individuals, especially, tossing a mask down the pit was worthwhile, even if it shortened the lifespan of the pit slightly: the perceived impact on the pit was small, they had clear alternatives at the end of its life, and this method of disposal was held to be the safest. Moreover, once the pit was covered, and the latrine moved, the discarded masks would be safely sealed away permanently.

## Guidelines, media, and common sense

In Malawi, official messaging around disposal has been mixed and behaviour has grown up as a common-sense response, rather than a reaction to specific state guidelines. Respondents described being inundated by news and information about Covid-19 from a diverse range of sources: official media in print, on T.V., visits community health workers, and the most digested medium, radio. These official channels of information run parallel to other sources of informal news on Covid-19, including local gossip, shared between neighbours, co-workers and at church, and information shared on social media platforms, predominantly Facebook and Whatsapp. The power of social media to spread miss-information on Covid-19 has already been commented, explicitly in Malawi [39]. Regardless, the cacophony of information (often conflicting) around the disease contributed to a degree of scepticism and mistrust amongst Blantyre respondents. One individual in Ndirande (07/10/2020) articulated this feeling:

> We used to trust the information, but now we don't really trust it, because most people these days don't believe that there is corona; maybe there is corona, but I doubt it. . . . I can`t believe what people are saying because it`s mixed information; some say it's real and some don't believe it exists. . . . [but what] you see on TV, the way doctor and nurses dress when handling a corona patient- it makes you believe.

As a result, respondents largely adapted what they considered to be their own best practice for daily living, and specifically disposal.

As previously noted, respondents overwhelmingly described disposing their masks down pit latrines, or burning them when possible. When pressed by the researchers about where they had learned these disposal methods, nearly all respondents replied that it was the guidance of the state (primarily conveyed through radio and the visits of community health workers). However, when pressed on this point it emerged that this was not exactly true, but that the guidance the state had been delivering was for citizens to dispose of masks somewhere safe, where they will not come into contact with humans. The state was not explicitly advocating for burning and pit toilet tossing as a best practice, but that is how it was interpreted by most respondents. The following responses to the question 'where did you learn to do that?' illustrate this point:

> I heard on the radio that people need to dispose of masks in a manner that ensures no person comes into contact with them as they may carry coronavirus. So, people dispose of masks in the pit latrines (08/10/2020).

> We have never heard it anywhere, but we just think that the best place to dispose of used masks is in the pit latrines as they may be contaminated (07/10/2020).

*They say we should be disposing of masks in an appropriate place. But, they don't straight for-wardly say we should be disposing of masks in pit latrine (10/03/2021).*

*I think it`s one of those things we are doing using our own judgement because disposing of masks in latrines eliminates any chance of people coming into contact with contaminated masks. I don't think I have heard it on the radio, but I hear it from people . . . We have just formed that habit within ourselves, and I don't know where it initially came from (08/03/2021).*

*I have never heard anywhere saying that masks should be disposed of in latrines or bins. The president and health professionals don't say where we should be disposing of masks. I like this question a lot, it`s a good one. I'm going to keep this for Sunday church service–I will surely ask it. . . . . .Where are people supposed to be disposing of masks*? *If I go to QECH* [Queen Elizabeth Central Hospital, the largest public hospital in Malawi], *am I supposed to dispose of it there (09/10/2020)*?

Thus, in Blantyre, disposing of used masks down pit latrines was not a behaviour taught or advocated for explicitly by the state, but was rather adopted organically by individuals who understood it to be best practice, given the circumstances.

### "The pit latrine is meant for this type of trash"

These behaviours are not a response to Covid-19, but adaptions of already existing practices. As previously described, Roxburgh et al. [36], write about how for Malawian women, pit latrines already fill essential SWM roles. Specifically, they investigate disposal of feminine hygiene products and menstrual blood, which women dispose down pit latrines in 'emergency' situations to avoid possible stigma. This suggests that, to some degree, the pit latrine has *always been a central part* of low-income households' SWM systems. Indeed, data from this investigation suggests that this behaviour was broadly held amongst our respondents, for a wide range of feared or otherwise hazardous waste items, independent of Covid-19. As such, the integration of the pit latrine into respondents' household solid waste management practices during the Covid-19 pandemic should not be seen as ad-hoc, or a specific response to a new waste stream, but the extension of an existing practice to new circumstances.

When asked generally, what they toss down their pit latrines, most respondents stressed that they do not use the pit to dispose of common household waste items, like paper or plastic, because, as the previous section mentioned, they are aware of the impacts that it will have on the pit, as well as the expenses, in time and labour, of digging a new one. However, when pressed about specific items that they may dispose down the pit, many respondents described a number of other potentially revealing items. Some mentioned items, such as condoms, also described in other pit latrine literature [33,34] which could be disposed down pits to avoid personal embarrassment or stigma (i.e. so that the neighbours or other household members do not see them). As one respondent (09/10/2020) who described tossing pits down their pit said, "it is for privacy, you don't need everyone to know what you are doing." Other types of items 'hidden' down pits include waste items that might be considered a safety hazard if disposed openly, such as razor blades or broken glass. In addition, Roxburgh et al. [36] described that one of the reasons that women dispose of feminine hygiene products down pits is anxiety over them being possibly recovered and used for *ufiti* or traditional forms of witchcraft. The connection between bodily fluids, waste, and the pit latrines, was also articulated by a few respondents of this study who describe occasionally tossing personal effects, such as old clothes down the pit, so that they could not be used for witchcraft. This one exchange (09/03/2021) typifies these responses:

*Interviewer: What else do you dispose in the pit latrine?*

*Interviewee: Some old rags because you can't just dispose them of anyhow, people can use old rags to bewitch you.*

*Interviewer: How does it happen?*

*Interviewee: (Laughs) People can get your old piece of cloth and perform rituals on it to do you harm, so we do that to protect ourselves.*

In this regard, the pit latrine was once again considered a 'safe' place to dispose of a waste item the disposer feared being found.

Lastly, as we have described, respondents problematised discarded masks, because they were aware, to varying degrees, that they could be a possible source of infection, and protecting themselves and their community from this source of infectious waste by disposing of it in an accessible location was one of the main motivators driving their disposal down the pit latrine. However, this is not a new behaviour, and seems to have been an established practice among respondents to use the pit latrine to dispose of possible infectious waste, or waste otherwise associated with sickness or disease. As one respondent (07/10/2020) candidly pointed out, "people are used to this. . . you have forgotten, we have other diseases [in our community]." As such, respondents described the pit as the routine and ideal place for such waste, including prior to the pandemic. In addition to tissues, bandages, needles, and items considered as more conventional medical waste, respondents also described using the pit to dispose of waste which they feared might cause other illnesses, such as dead animals, which can safely decompose in the pit, away from the community. Moreover, we believe these practices and beliefs to be broadly held. When asked how these habits had formed, most respondents expressed that that is how it had always been done, and it was not the advice of the state or community health workers. As one individual in Ndirande (10/03/2021) described, "everyone knows to do this without being told–It's just basic. The pit latrine is meant for this type of waste."

## Discussion

In addition to its human cost, the still on-going global Covid-19 pandemic has taken an immense toll on nearly every facet of daily life. Moreover, it has created, and continues to create, a vast amount of solid waste, which necessitates efficient and sanitary management and disposal to stem the spread of the disease. However, throughout the Global South, and in Africa in particular, large segments of the population lack access to reliable municipal solid waste management services, complicating the disposal of infectious waste fractions, and thus, requiring individual households to devise their own 'safe' disposal methods. Drawing on extensive qualitative fieldwork, including 96 semi-structured interviews, across four different low-income communities in Blantyre, Malawi and Durban, South Africa, the purpose of this article was to respond to a qualitative gap on mask disposal behaviours, particularly from within low-income and African contexts. Specifically, our purpose was to understand what behaviours have arisen over the past year, across the two disparate national contexts, and how they have been influenced by individual risk perceptions, established traditional practice, state communication, and other media sources.

The case study of Durban has been primarily useful as a contrast to that of Blantyre. Although all of the study's respondents were purposively selected from similarly low-income communities, Durban respondents exhibited dramatically different masking and mask disposal behaviours. Residents of Johanna Road overwhelmingly utilised cloth masks, which they washed and reused daily. Thus, for them, disposal was not an issue in their lives. Moreover,

although Johanna Road is an informal settlement, it enjoys significantly better public services than Ndirande and Likhubula, including reliable municipal solid waste collection, which was not seriously disrupted during the pandemic. So, for the few Durban respondents that did dispose of their masks, they were able to do so efficiently with their bagged household waste. Surely much is still to be written on the efficacy of different mask typologies, and there already has been some scholarship on the value of cloth masks in Global South spaces [29,40] however, from a waste management perspective, this study suggests that more widespread usage of cloth face coverings may help to reduce the waste management and disposal burden, especially in the short term, while minimising localised risk from discarded disposable masks.

Across the different Blantyre neighbourhoods, respondents more commonly used disposable masks. Moreover, proper disposal was a priority, as most respondents had a clear understanding that a used mask was harmful; as something infectious that could possibly transmit Covid-19 to the vulnerable with their community, such as children. However, these communities also lack reliable waste management services, with open dumping and household burning being the most common means of disposing of household waste, with open dumping not being considered as a 'safe' way to dispose of a used mask, as they could be picked up out of the dump piles by children, while burning was considered ideal, but laborious and inconvenient. As such, more than 90 percent of Blantyre respondents reported using a pit latrine to dispose of their mask, at least some of the time, and the pit latrine was characterised as the ultimate 'safe' place to dispose of a mask, as once dropped, it was gone for good. However, the pit latrine has not been an ad-hoc integration into individual's household solid waste management system during the pandemic—it was *already* an essential part of it. Lacking reliable municipal services, respondents demonstrated an established practice of using the pit latrine to dispose of embarrassing, dangerous, or possibly infectious waste items, which they could not otherwise safely or discreetly dispose of within their community. As such, within these contexts, the pit latrine fulfils a valuable and underreported solid waste management function, in addition to its sanitation role.

## Conclusion

This study has a number of implications for both SWM and WASH interventions in African, and Global South contexts, more broadly. First, and most importantly this work highlights the inextricable link between sanitation technologies for excreta and solid waste management. Though known implicitly by WASH practitioners for decades, we have seen few instances of these linkages being formalised either through research, funding, or education. Yet, the sanitation sector, particularly the part focused on pit emptying, has remained surprisingly resigned to deal with the physical manifestations of the SWM sector's inability to collect and manage waste. The current push to valorise faecal sludge for energy, nutrients, and protein [41–43] cannot practically succeed without a trash-free faecal sludge feedstock. Engineers, designers, and policy makers have developed and implemented numerous innovative sanitation technologies over the last 20+ years, but never with the consideration of trash; all of these technologies have seemingly been designed in a solid waste free bubble; designed to work in perfect conditions, conditions which we, and other literature [36] have demonstrated, simply do not exist.

Second, these findings suggest that a paradigm shift is needed, away from technologies and practices which ignore that pits are used for solid waste, towards approaches which acknowledge and value the undercounted solid waste management function of this infrastructure. The Covid-19 pandemic has caused enumerable disruptions to service delivery across the globe, yet it has also underscored the latent inequality that already existed between the rich and the poor, between the North and the South, in access to reliable and safe solid waste management

services [39]. During a global pandemic, disposal of possibly infectious waste down a pit latrine, in such contexts, may not be considered just a viable alternative to dumping or burning, but best practice. Moreover, if this investigation has demonstrated anything, it is that it is misguided to chastise the toilet owners to use their latrine 'improperly' despite having no safe alternative to manage their solid waste. These findings are a call to the WASH and particularly the Faecal Sludge Management (FSM) community that the technologies and the business models of the future must necessarily consider the presence of these masks, and realistically, a variety of trash items, not just post-Covid, but in all urban centres with limited SWM, where citizens have come to rely on the humble pit as their most trusted trash receptacle.

## Acknowledgments

The authors would like to thank Green Corridors for supporting the fieldwork in Durban. The authors would also like to thank and acknowledge the feedback of the reviewers in contributing to this manuscript's development.

## Author Contributions

**Conceptualization:** Marc Kalina, Elizabeth Tilley.

**Data curation:** Marc Kalina, Jonathan Kwangulero.

**Formal analysis:** Marc Kalina.

**Funding acquisition:** Elizabeth Tilley.

**Investigation:** Marc Kalina, Jonathan Kwangulero, Fathima Ali.

**Methodology:** Marc Kalina.

**Project administration:** Marc Kalina.

**Software:** Marc Kalina.

**Supervision:** Marc Kalina, Fathima Ali, Elizabeth Tilley.

**Validation:** Marc Kalina.

**Visualization:** Marc Kalina, Elizabeth Tilley.

**Writing – original draft:** Marc Kalina, Elizabeth Tilley.

**Writing – review & editing:** Marc Kalina, Elizabeth Tilley.

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
