## [Decision Letter · Decision Letter 0]

2 Sep 2021

PONE-D-21-20617

“You Need to Dispose of them Somewhere Safe”: Covid-19, Masks, and the Pit Latrine in Malawi and South Africa

PLOS ONE

Dear Dr. Mark Kalina,

Thank you for submitting your manuscript to PLOS ONE. After careful consideration, we feel that it has merit but does not fully meet PLOS ONE’s publication criteria as it currently stands. Therefore, we invite you to submit a revised version of the manuscript that addresses the points raised during the review process.

We look forward to receiving your revised manuscript.

Kind regards,

Balasubramani Ravindran, Ph.D

Academic Editor

PLOS ONE

Journal Requirements:

2. We note that Figure(s) 1 and 2 in your submission contain [map/satellite] images which may be copyrighted. All PLOS content is published under the Creative Commons Attribution License (CC BY 4.0), which means that the manuscript, images, and Supporting Information files will be freely available online, and any third party is permitted to access, download, copy, distribute, and use these materials in any way, even commercially, with proper attribution. For these reasons, we cannot publish previously copyrighted maps or satellite images created using proprietary data, such as Google software (Google Maps, Street View, and Earth). For more information, see our copyright guidelines: http://journals.plos.org/plosone/s/licenses-and-copyright.

1. You may seek permission from the original copyright holder of Figure(s) 1 and 2 to publish the content specifically under the CC BY 4.0 license.  

Reviewers' comments:

Reviewer's Responses to Questions

**Comments to the Author**

1. Is the manuscript technically sound, and do the data support the conclusions?

Reviewer #1: Yes

Reviewer #2: Yes

2. Has the statistical analysis been performed appropriately and rigorously? 

Reviewer #1: N/A

Reviewer #2: N/A

3. Have the authors made all data underlying the findings in their manuscript fully available?

Reviewer #1: Yes

Reviewer #2: No

4. Is the manuscript presented in an intelligible fashion and written in standard English?

Reviewer #1: Yes

Reviewer #2: Yes

5. Review Comments to the Author

Reviewer #1: The manuscript was is articulated and presented. However, ethical issues was not mentioned under the methodology as well as informed consent by the participants. The manuscript is presented in an intelligible fashion and in standard English.

Reviewer #2: The manuscript is good and the interviews were appropriate. Authors should include a paragraph summarising the transmission and of coronaviruses via face masks and other means of transmission of the virus from person to person. Also, discussion and conclusion should be revised appropriately.

6. PLOS authors have the option to publish the peer review history of their article (what does this mean?). If published, this will include your full peer review and any attached files.

Reviewer #1: **Yes: **OLUSESAN ADEYEMI ADELABU

Reviewer #2: No

---

## [Author Response · Author response to Decision Letter 0]

3 Sep 2021

To the editor and reviewers, 

We want to thank everyone for their time and feedback. We have engaged broadly with the comments and integrated them as thoroughly as possible. Regarding the specific formatting queries from editor, we have revised the manuscript to be compliant with the Journal's formatting requirements, and have removed the maps, as licensing seemed to be an onerous process and we do not feel they were essential to the manuscript. The supporting information file (interview schedules) was only for review and has not been included in this submission.

Regarding the review comments, thank you again to both reviewers for your time and comments. We have taken the them time to engage with them as thoroughly as possible. Please see our attached responses for a list of specific actions taken.

Thank you.

---

## [Decision Letter · Decision Letter 1]

5 Jan 2022

“You Need to Dispose of them Somewhere Safe”: Covid-19, Masks, and the Pit Latrine in Malawi and South Africa

PONE-D-21-20617R1

Dear Dr. Marc Kalina,

We’re pleased to inform you that your manuscript has been judged scientifically suitable for publication and will be formally accepted for publication once it meets all outstanding technical requirements.

Kind regards,

Balasubramani Ravindran, Ph.D

Academic Editor

PLOS ONE

Reviewers' comments:

Reviewer's Responses to Questions

**Comments to the Author**

1. If the authors have adequately addressed your comments raised in a previous round of review and you feel that this manuscript is now acceptable for publication, you may indicate that here to bypass the “Comments to the Author” section, enter your conflict of interest statement in the “Confidential to Editor” section, and submit your "Accept" recommendation.

Reviewer #2: All comments have been addressed

2. Is the manuscript technically sound, and do the data support the conclusions?

Reviewer #2: Yes

3. Has the statistical analysis been performed appropriately and rigorously? 

Reviewer #2: N/A

4. Have the authors made all data underlying the findings in their manuscript fully available?

Reviewer #2: No

5. Is the manuscript presented in an intelligible fashion and written in standard English?

Reviewer #2: Yes

6. Review Comments to the Author

Reviewer #2: The manuscript is a good one and it will interest the readers. I recommend the publication of the manuscript after addressing the minor comments suggested.

7. PLOS authors have the option to publish the peer review history of their article (what does this mean?). If published, this will include your full peer review and any attached files.

Reviewer #2: No

---

## [Editor Report · Acceptance letter]

3 Feb 2022

PONE-D-21-20617R1 

“You Need to Dispose of them Somewhere Safe”: Covid-19, Masks, and the Pit Latrine in Malawi and South Africa 

Dear Dr. Kalina:

I'm pleased to inform you that your manuscript has been deemed suitable for publication in PLOS ONE. Congratulations! Your manuscript is now with our production department. 

Kind regards, 

on behalf of

Dr. Balasubramani Ravindran 

Academic Editor

PLOS ONE